# Emptying Dirac valleys in bismuth using high magnetic fields

Zengwei Zhu[1,2], Jinhua Wang[1], Huakun Zuo[1], Benoît Fauqué[3,4], Ross D. McDonald[2], Yuki Fuseya[5] & Kamran Behnia[1,3]

The Fermi surface of elemental bismuth consists of three small rotationally equivalent electron pockets, offering a valley degree of freedom to charge carriers. A relatively small magnetic field can confine electrons to their lowest Landau level. This is the quantum limit attained in other dilute metals upon application of sufficiently strong magnetic field. Here we report on the observation of another threshold magnetic field never encountered before in any other solid. Above this field, $B_{empty}$, one or two valleys become totally empty. Drying up a Fermi sea by magnetic field in the Brillouin zone leads to a manyfold enhancement in electric conductance. We trace the origin of the large drop in magnetoresistance across $B_{empty}$ to transfer of carriers between valleys with highly anisotropic mobilities. The non-interacting picture of electrons with field-dependent mobility explains most results but the Coulomb interaction may play a role in shaping the fine details.

[1] Wuhan National High Magnetic Field Center and School of Physics, Huazhong University of Science and Technology, Wuhan 430074, China. [2] MS-E536, NHMFL, Los Alamos National Laboratory, Los Alamos, New Mexico 87545, USA. [3] Laboratoire Physique et Etude de Matériaux (CNRS-UPMC) ESPCI Paris, PSL Research University, Paris 75005, France. [4] JEIP, USR 3573 CNRS, Collège de France, PSL Research University, 11, place Marcelin Berthelot, Paris Cedex 05 75231, France. [5] Department of Engineering Science, University of Electro-Communications, Chofu, Tokyo 182-8585, Japan. Correspondence and requests for materials should be addressed to Z.Z. (email: zengwei.zhu@hust.edu.cn).

Similar to spin and unlike momentum, the valley degree of freedom is a discrete quantum number for Bloch waves in a crystal, with a potential for storing information. Valleytronics[1] has been initially explored in AlAs-based two-dimensional electron gas[2] and subsequently in a variety of systems such as graphene[3], transition-metal dichalcogenides[4], diamond[5] and silicon[6]. Magnetic field can modulate occupation of different valleys[7,8] or their contribution to the total conductivity[9–11]. In bismuth, the latter effect is visible even at room temperature and in the presence of a field as weak as 0.5 T (refs 9,10).

The large orbital magnetoresistance of bismuth has been known for several decades[12,13]. Following the recent observation of a large magnetoresistance in WTe$_2$[14], the amplitude of magnetoresistance in compensated semi-metals has attracted renewed attention[14–17]. These studies have brought to foreground a fundamental yet unanswered questions: what sets the magnitude and field dependence of orbital magnetoresistance in the high-field regime where the scattering time exceeds by far the cyclotron period[18,19]?

A longstanding open question in condensed-matter physics is the fate of the three-dimensional electron gas pushed beyond the quantum limit[20]. When electrons are confined to their lowest Landau levels, the Coulomb interaction, neglected in the single-particle treatment of electrons, is expected to play a significant role. While many dilute metals are pushed beyond the quantum limit, graphite remains the only 3D system in which electronic instabilities[21], believed to be charge–density–wave states[22], have been detected. A unified picture of this regime and the consequences of multiple valleys are yet to be drawn.

We present a study of magnetoresistance of bismuth up to 90 T, and a comprehensive angle dependence up to 65 T, with implications for our understanding of magnetoresistance in a multi-valley electron system far beyond the quantum limit. Previous studies demonstrated that by lifting the degeneracy of anisotropic valleys, magnetic field affects their contribution to the total conductivity[9–11]. When the quantum limit is approached, even the carrier population in each valley becomes different[23]. However, prior to these measurements magnetic fields high enough to wipe out a valley evaporating its Fermi sea had not been applied. Our study detects a large anisotropic drop in magnetoresistance. Comparing the data with theoretical calculations, we argue that this drop happens because one or two Fermi seas, depending on the orientation of magnetic field, totally dry up. Thus bismuth becomes the only solid in which the amplitude of the magnetic field to attain 100% valley polarization is known and attainable. The semi-classic and multi-valley[24] picture of non-interacting electrons provides a qualitative explanation of the observation. The large drop in magnetoresistance is a consequence of transfer of carriers from a high-mobility to a low-mobility valley. A quantitative model for the drop of resistance at $B_{empty}$ would require a full consideration of the role of Coulomb interactions.

## Results

**The three valleys**. As reminded in Fig. 1a–c, bismuth is a compensated semi-metal with electrons at L-point on top of a very small gap and with a quasi-linear dispersion, giving rise to three Dirac valleys, which are elongated Fermi surface ellipsoids located at the boundary of the Brillouin zone (For a review, see ref. 25). The extreme anisotropy of these ellipsoids make them quasi-two-dimensional (Q2D) metals. In real space, Bloch waves of each three Q2D metal can be easily displaced by electric field along the trigonal axis and one of the three binary axes, but not along the third perpendicular orientation, one of the three bisectrix axes. The rhombohedral structure of the bismuth crystal

can be seen as a distortion of two interpenetrating face-centred cubic (FCC) structures (Fig. 1c). For each valley index, the preferential charge flow along a binary axis favours one out of the three bonds linking atoms of one FCC network (in yellow) to two of its nearest neighbours belonging to the other FCC network (in black)(Fig. 1c). According to the present findings, a sufficiently strong magnetic field can completely freeze one or two of these three equivalent channels.

**Angle-dependent magnetoresistance**. Our principal experimental result is presented in Fig. 1d. For both binary and bisectrix orientations of magnetic field, the magnetoresistance stops to increase and begins to drop when the field $>35$ T. When the field is along the binary axis, the drop is more drastic and starts at a slightly higher magnetic field. As we will see below, both these features find a natural explanation if the drop is due to the total evacuation of one or two electron pockets. The angle dependence of the magnetoresistance presented in a polar plot in Fig. 1e clarifies the link between this result and previous studies of angle-dependent magnetoresistance in bismuth[9,10]. As seen in the figure, at 5 T, angular oscillations of magnetoresistance are visible, arising from the anisotropy of the mobility. For each ellipsoid, the magnetoresistance is larger when the field is along its longer axis. As the field increases, the relative magnitude of the oscillations are damped because the relative contribution of the hole pocket to the total conductivity is enhanced. At 55 T, however, pronounced angular oscillations appear again with an amplitude much larger than what seen at low fields. Note also the loss of threefold symmetry[9,10,26], which leads to the slight but visible different in $\rho(B)$ for nominally equivalent binary and bisectrix axes. The origin of this field-induced 'nematicty' is a subject of ongoing research. We also measured magnetoresistance of another bismuth crystal up to 90.5 T with magnetic field oriented along the binary axis and found that the system remains metallic with significant implications for the Landau sepctrum as discussed below.

**Theoretical Landau spectrum**. Before discussing the origin of the drop in magnetoresistance, let us consider the Landau spectrum of bismuth in the present configuration. Our data extends the maps of Landau spectrum in bismuth previously drawn up to 12 T (ref. 23) and 28 T (ref. 27) based on studies of the Nernst effect, to 65 T. Figure 2 shows the theoretical evolution of the Landau levels, the Fermi energy (Fig. 2a,d), the carrier concentration (Fig. 2b,e) and its distribution among valleys (Fig. 2c,f) with magnetic field (see more about the theoretical details in Supplementary Note 1). With magnetic field perpendicular to the trigonal axis and $>15$ T, electrons are confined to the lowest spin-polarized Landau sub-level ($0_{e-}$ ref. 23) and holes occupy their three (each doubly degenerate) lowest Landau levels. With increasing magnetic field, three events occur at successive magnetic fields. The $2_h$ and $1_h$ Landau levels cross the Fermi level and last occupied electron LL for one (when $B\|$bisectrix) or two (when $B\|$binary) pockets becomes empty. The Fermi energy is significantly affected by a modest magnetic field. This is because the carriers are light (almost a thousandth of the free electron mass along bisectrix[23]) and therefore the cyclotron energy is large. Charge neutrality imposes equality between electron and hole concentrations. Since electron degeneracy rises faster than hole degeneracy, the Fermi energy shifts downward with magnetic field up to 20 T. The evacuation of each hole Landau level produces a deceleration kink in this downward shift. Note the non-monotonic field dependence of the $0_{e-}$ level for different valleys. It depends on the magnitude of the interband coupling and the g-factor corrections (see more about the theoretical

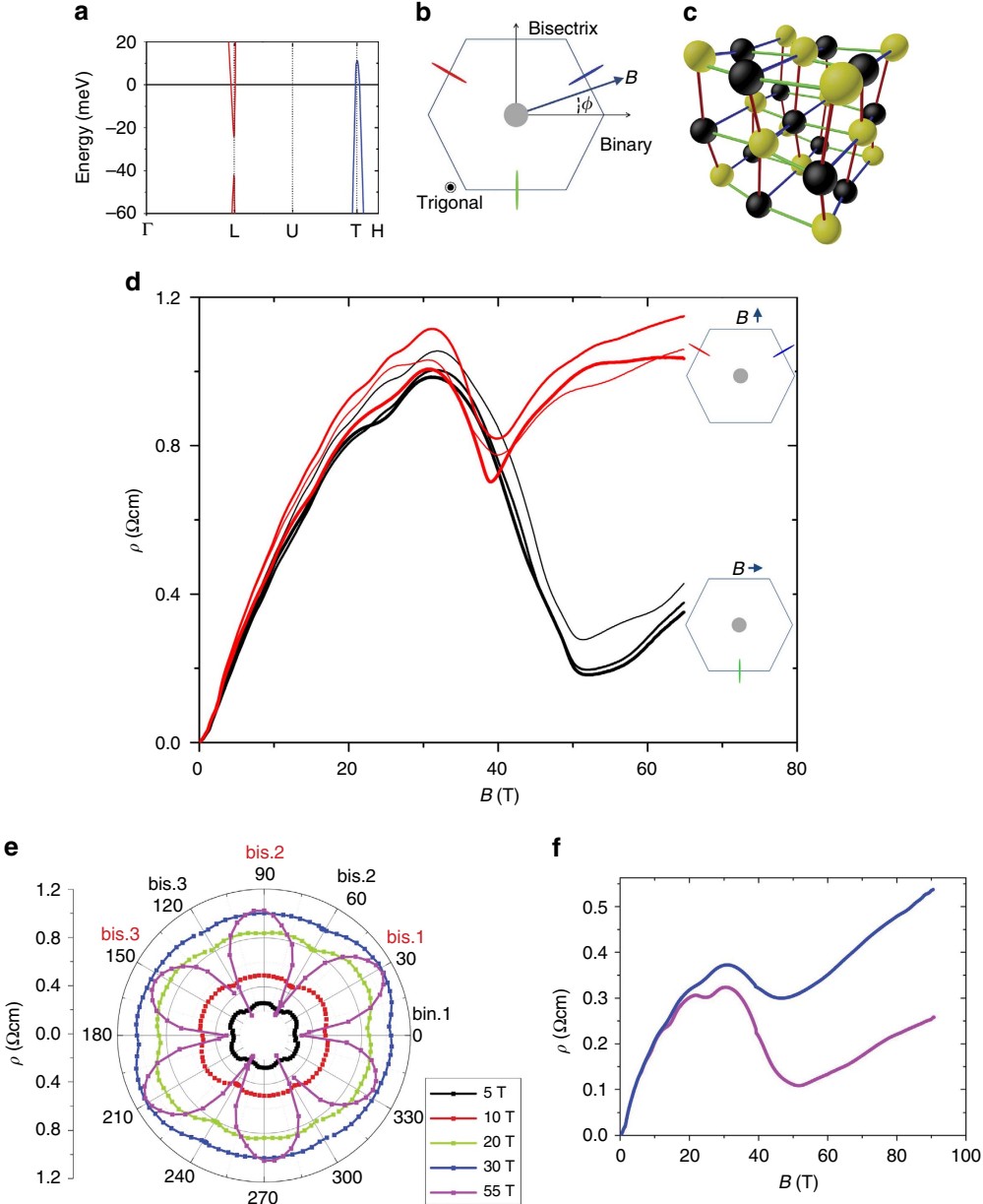

**Figure 1 | Bismuth structure and its high-field magnetoresistance. (a)** Dirac-like dispersion of electrons near L-point and parabolic dispersion of holes near T-point of the Brillouin zone in bismuth. The horizontal axis is for the wave vector along the high symmetry points. **(b)** Dirac valleys are elongated ellipsoids at the boundaries of the Brillouin zone. $\phi$ is the angle between the magnetic field and one of the binary axes. The central grey circle is for the hole pocket while the electron pockets are in red, blue and green, which corresponds to the bands with same colours in **c**. **(c)** Lattice structure seen as two interpenetrating face-centred-cubic (FCC) sub-lattices. In real space, each valley is associated with easy flow of charge between neighbouring atoms, indicated by the rod colours. **(d)** Transverse magnetoresistance of a bismuth crystal up to 65 T at $T = 1.56$ K for magnetic fields along the three equivalent binary (black) and bisectrix (red) crystalline axes labelled in **e**. Thicker lines correspond to larger $\phi$. The large drop is a consequence of the total evacuation of one or two electron valleys as sketched in the inset. **(e)** Polar plot of angle $\phi$ dependence of magnetoresistance for different fields at $T = 1.56$ K. Note the dramatic enhancement of angular oscillations at 55 T. The abbreviation bin. (bis.) denotes binary (bisectrix) axes. **(f)** Magnetoresistance up to 90.5 T in another bismuth single crystal at 1.4 K (magenta) and 9 K (blue) as the field is along binary and the current is along bisectrix. The system remains metallic in the whole field range.

details in Supplementary Note 1) and sets the field at which a valley becomes empty.

As seen in Fig. 2b,e, the carrier concentration is drastically modified by magnetic field. It increases first, before saturating and dropping at higher fields. The distribution of carriers among electron pockets (Fig. 2c,f) is also strongly affected by magnetic field. It ends up by total evacuation of one or two electron valleys. Thus magnetic field can weigh enough on the balance of energy to dry a Fermi sea without causing a metal–insulator transition. The

total number of remaining electrons is still matching the number of holes respecting charge neutrality and keeping the system metallic.

**Experimental Landau spectrum**. Figure 3a presents a colour map of the angle-dependent magnetoresistance. The angular evolution of the second derivative is presented in Fig. 3b,c. The evacuation of each hole Landau level occurs at a field that does not strongly vary much with azimuthal angle. The Zeeman splitting of the hole

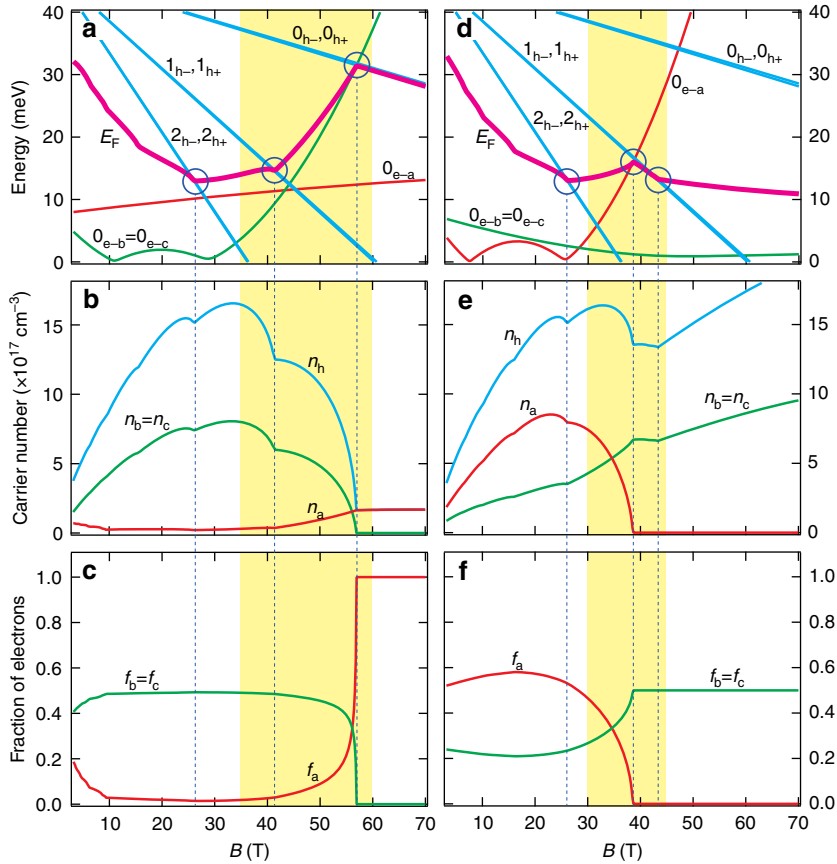

**Figure 2 | Theoretical Landau spectrum. (a,d)** Field dependence of the Landau levels and the Fermi energy and **(b,e)** carrier concentration for $B\|$binary **(a–c)** and $B\|$bisectrix **(d–f)**. a, b and c indices refer to the three electron pockets and h to the hole pocket. **(c,f)** The proportion of carriers in different electron pockets as a function of magnetic field. The shaded region in yellow marks the boundaries of the experimentally observed magnetoresistance drop. It matches the theoretical field window for carrier transfer between valleys. The vertical dashed lines mark the magnetic fields at which a Landau level is evacuated by crossing the Fermi level.

Landau levels vanishes when the field is perfectly perpendicular to the trigonal axis[28] but becomes finite in case of slight misalignment. With these features in mind, one can identify the horizontal bright lines of Fig. 3b as hole Landau levels and their bifurcation at intermediate angles as the result of finite Zeeman splitting arising from imperfect alignment. The experimental data in Fig. 3c is to be compared with the theoretical angle-dependent Landau spectrum of Fig. 3d (which takes into account the misalignment; see more about the theoretical details in Supplementary Note 1). The excellent agreement validates parameters used for the simulation. The drop in magnetoresistance occurs when the $0_{e-}$ electron Landau level of a valley crosses the chemical potential, leaving no carriers in the valley.

## Discussion

The large drop in magnetoresistance cannot be due to the evacuation of the $1_h$ hole level, as suggested decades ago by authors observing the onset of this drop[29,30] and using a theoretical model contradicted by the absence of metal–insulator transition evidenced by Fig. 1f (see more about the theoretical details in Supplementary Note 1 for details). This interpretation becomes implausible due to the sharp angular dependence of the drop resolved here. Why is the drop much larger than any other quantum oscillation arising from the evacuation of a hole or electron Landau level (Fig. 1d,f) and has a different temperature dependence (see the experimental results in Supplementary Note 2)? Why it is more than twice larger and significantly wider

when the field is along the binary axis? Such questions find straightforward answers as soon as one considers the consequences of emptying an electron valley by magnetic field.

As seen in Fig. 1d, magnetoresistance in bismuth is approximately linear in field over an extended window of magnetic field up to 25 T. Its angle dependence in the presence of moderate magnetic field can be quantitatively explained in a semi-classical approach taking carrier mobility as a tensor[10]. The non-trivial field dependence of magnetoresistance (its deviation from the quadratic behaviour expected for a compensated semi-metal) reflects the fact that both carrier density and the components of the mobility tensor vary with magnetic field[13,18]. In this context, a change in total carrier concentration, a change in the occupation of different valleys or a change in the relevant mobility component are three distinct possibilities to cause a change in magnetoresistance. Let us examine each of these possibilities.

As seen in Fig. 2, following the evacuation of $1_h$ Landau level, carrier concentration decreases. This would lead to an enhancement (and not a decrease) in magnetoresistance. Whatever causes the drop in magnetoresistance should counter and outweigh the enhancement due to the decreased carrier number. On the other hand, the change in the fraction of electrons occupying each valley (shown in Fig. 2c,f) would generate a drop in magnetoresistance according to what we know of the mobility anisotropy of each valley. When the field is parallel to the longer axis of one electron ellipsoid, the relevant component of the mobility tensor for electrons in this ellipsoid is large.

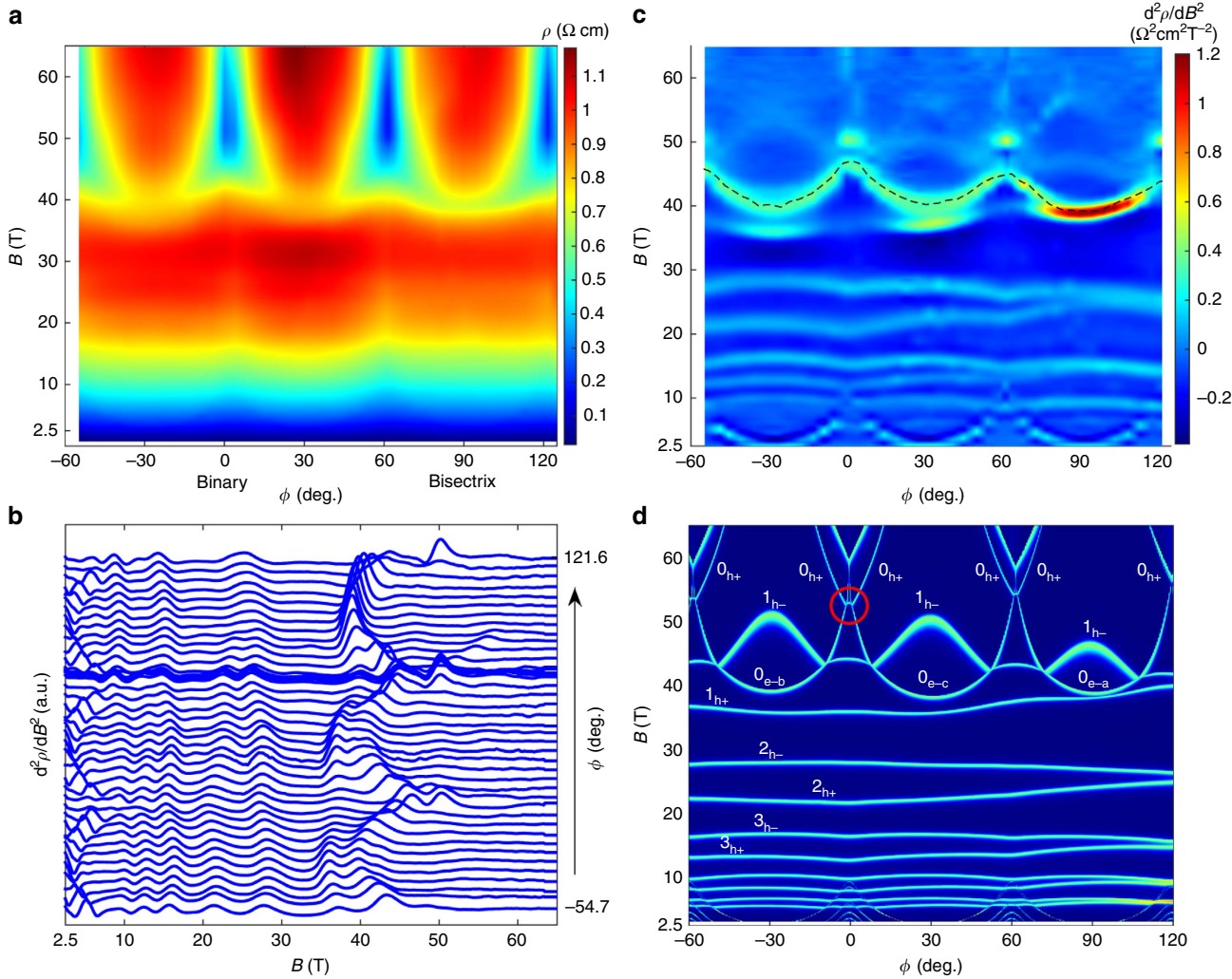

**Figure 3 | Comparison of theoretical and experimental Landau spectrum.** (**a**) Colour plot of magnetoresistance. The large drop in magnetoresistance is visible as blue regions at high field near the binary axis. (**b**) Angular evolution of the second derivative of magnetoresistivity, $d^2\rho/dB^2$. Peaks are caused each time a Landau level crosses the Fermi level. (**c**) Colour plot of the same data reveals the angle-dependent Landau spectrum. Bright lines corresponding to the evacuation of electron and hole Landau levels are identified. The $0_{e-}$ level is marked with a black dash line. (**d**) Plot of the theoretical Landau spectrum assuming a small misalignment. The bright lines are the Landau levels for electrons (e) and holes (h), some of which are indexed. When the $0_{e-}$ level of a valley crosses the Fermi level, it becomes empty. Note that the blue regions in **a** coincide with the region in **d** (highlighted with a circle), where three Landau levels (two electron and one hole) simultaneously cross the chemical potential.

When this valley is emptied and its electrons move to the other two, the relevant component of the mobility tensor becomes lower and therefore magnetoresistance drops. When the magnetic field is along the binary axis, two valleys become empty with consequences more drastic than emptying one, hence a larger drop in magnetoresistance. As seen in Fig. 2c,f, the field window for carrier transfer between valleys is remarkably close to the width of the experimentally observed drop in magnetoresistance.

We can express our interpretation in quantitative terms. In the high-field limit ($\mu B \gg 1$), the conductivity for charge current applied along the trigonal axis, $\sigma_{33}$, can be approximated in the following way (see the detailed derivation in Supplementary Note 3):

$$\sigma_{33}^{\text{bin}} \simeq \frac{en_h}{B^2}\left(\frac{f_a}{\mu_a} + 2\frac{f_b}{\mu_b}\right) \quad (1)$$

$$\sigma_{33}^{\text{bis}} \simeq \frac{en_h}{B^2}\left(\frac{f_a}{\mu_a'} + 2\frac{f_b}{\mu_b'}\right) \quad (2)$$

The indexes bin and bis refer to orientation of the magnetic field. $n_h$, $n_a$, $n_b$ and $n_c$ are carrier densities of the hole pocket and

the electron pockets $a$, $b$ and $c$ (for $B\|$binary and $B\|$bisectrix, $n_b = n_c$). $f_a = \frac{n_a}{n_a + 2n_b}$, $f_b = \frac{n_b}{n_a + 2n_b}$ are the fraction of carriers residing in a valley. Parameters $\mu_a$, $\mu_b$, $\mu_a'$ and $\mu_b'$ are effective mobilities, each combinations of the components of the electron and hole mobility tensor (see the detailed analysis in Supplementary Note 3). If we assume that the effective mobilities remain constant, the change in conductivity depends on the variation of $n_h$, $f_a$ and $f_b$. The decrease in $n_h$ pulls down conductivity. But since $\mu_a < \mu_b$ and $\mu_a' > \mu_b'$, the change in $f_a$ and $f_b$ leads to an opposite effect. Because one or two valleys are emptied, $f_a$ and $f_b$ change more drastically than $n_h$ (see Fig. 2). Therefore, the increase in conductivity caused by the transfer of carriers from one valley to another easily outweighs the decrease expected due to the change in total carrier density.

We note that the effect discussed here bears a similarity to the Gunn effect[31]. In a semiconductor such as GaAs, a sufficiently strong electric field can displace carriers from one band to another causing a change in differential conductivity[32]. The effect we are observing here is also a consequence of carrier transfer

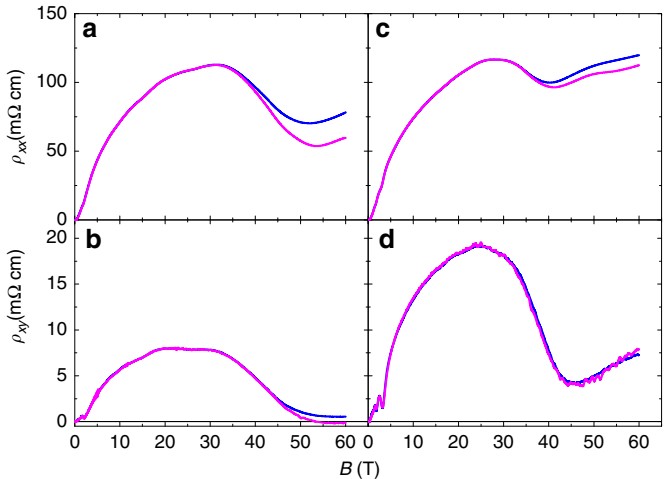

**Figure 4 | Hall data.** Longitudinal and Hall resistivity in another bismuth single crystal when the current is along trigonal as (**a**,**b**) for *B*||binary and the (**c**,**d**) for *B*||bisectrix at 1.66 K (magenta) and 4.2 K (blue). For both orientations of magnetic field, the Hall resistance shows a large drop in the vicinity of 40 T. For both orientations, the change in the Hall response is more drastic than the change in longitudinal resistance.

between branches with different mobilities caused by magnetic (and not electric) field.

We also measured the variation of the Hall resistivity, $\rho_{xy}$, up to 60 T. The results are presented in Fig. 4. As seen in the figure, for both configurations, $\rho_{xy}$ shows a jump more drastic than the one seen in the longitudinal resistivity, $\rho_{xx}$. In a compensated semi-metal, a finite Hall resistivity is expected if the mobility of hole-like and electron-like carriers differ, which is the case of bismuth. In semi-classical picture of multi-valley transport[24], the Hall conductivity is expected to vanish in the high-field limit ($\mu_i B \gg 1$) when the current is injected along symmetry axes. However, this picture assumes that carriers are not scattered from one valley to another. The finite Hall conductivity observed here indicates the existence of such scattering events. Interestingly, the Hall response becomes vanishingly small above $B_{empty}$ for the *B*||binary configuration (and not for *B*||bisectrix). Only in the former configuration electrons are confined to a single valley. Therefore, the magnitude of $\rho_{xy}$ at 60 T for the two configurations is in qualitative agreement with our picture.

A quantitative explanation of the magnitude of the drops in $\rho_{xx}$ and $\rho_{xy}$ requires an accurate knowledge of the evolution of the components of the mobility tensor across $B_{empty}$. In Supplementary Note 4, we employ the semi-classical transport theory to fit the experimental data over a wide field window. This narrows down the possible scenarios for the evolution of mobility with magnetic field and leads us to conclude that the non-interacting picture gives a reasonable account of the data. However, the absence of a perfect fit leaves room for a possible role of Coulomb interaction. The sharpness, the amplitude and the anisotropy of the drop in magnetoresistance cannot be reproduced without invoking an abrupt change (by roughly a factor of two) in $\mu_1$ and $\mu_2$, the components of the electron mobility tensor in the vicinity at $B_{empty}$. This leaves room for speculation on the role played by interaction.

What can cause such a sudden change in mobility in the vicinity of $B_{empty}$? We note that there is a correlation between the boundaries of the triangular blue regions of Fig. 3a and the contours of Landau levels in Fig. 3d. As highlighted by a red circle, two electron lines and one hole line meet together where the drop in magnetoresistance occurs and emphasize that such a meeting never occurs for Landau levels with a higher index

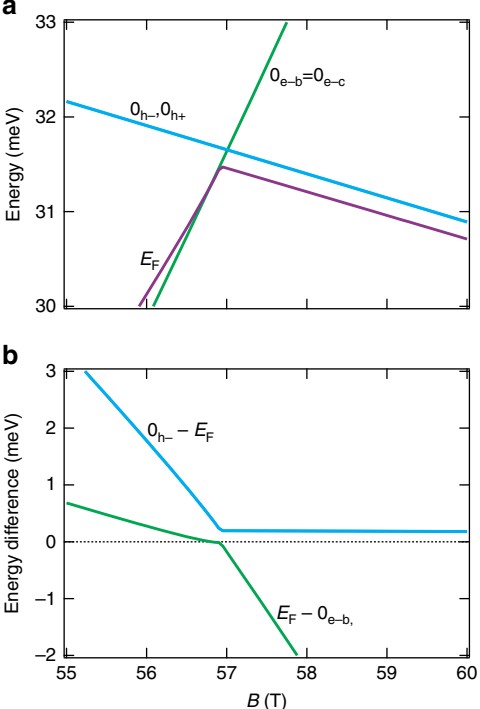

**Figure 5 | Zoom on theoretical spectrum near $B_{empty}$.** (**a**) Zoom on theoretical evolution of the Landau levels and the Fermi energy of Fig. 2a (*B*||Binary). This is where the two lowest electron Landau levels and the penultimate hole Landau level cross the Fermi level. (**b**) The difference of energy between electron and hole Landau levels and the Fermi energy. Bound excitonic states can be formed if the Coulomb attraction between electrons and holes exceed the sum of kinetic energy of electrons and holes. Note that here the calculations assume $M_{s3}^{-1} = 10^{-8}$. In principle, the Zeeman energy of holes becomes zero when the field is perpendicular to the trigonal axis[28] (see more description in Supplementary Note 1) and therefore $M_{s3}^{-1} = 0$.

number. When this meeting occurs (for $B \sim 50\,T$||binary), the energy cost of a trion (a charged exciton with two electrons and one hole), which is the kinetic energy of L-point electrons and T-point holes, becomes vanishingly small. Trions have been observed in semiconductors subject to frequency-tuned photons[33]. The formation of excitons in a semi-metal such as bismuth as a consequence of Coulomb attraction between electrons and holes is a longstanding idea[34]. The angle-dependent Landau spectrum emerging from this study specifies a region where their energy cost becomes very small. For *B*||binary, according to our calculations, the difference in energy between electron and hole Landau levels across the Fermi energy becomes as low as 0.5 meV (See Fig. 5). Now, with a mean distance between electrons and holes of $\ell_{eh} \sim 10\,nm$ and a dielectric coefficient of $\epsilon \sim 100\epsilon_0$, the exciton binding energy of an electron–hole pair is $E_{exc} = \frac{e^2}{4\pi\epsilon\ell_{eh}} \sim 0.7$ meV. At this stage, we can only speculate if it is more than accidental that the experimentally observed change in magnetoresistance occurs when the apparent energy cost of binding electrons to holes becomes comparable to the estimated magnitude of attraction between them.

In summary, based on an extensive set of angle-dependent magnetoresistance data, we conclude that the drop in magnetoresistance is not due to the evacuation of a hole Landau level as proposed previously[29,30], but the consequence of emptying at least one electronic valley. Thus our study identifies bismuth as the first solid in which cyclotron energy is strong enough to dry a Fermi sea. This happens because of a unique combination of

several features. First of all, the system is dilute and therefore the Fermi energy is small. Second, the carriers are light and both the cyclotron and the Zeeman energy are large. Third, unlike graphene, bismuth is not a perfect Dirac system. In a perfect Dirac system, the spin is locked to angular momentum leading to a cancellation of field-induced cyclotron and Zeeman shifts. Therefore, the lowest Landau level does not shift with magnetic field. The non-trivial evolution of the lowest Landau level (see more about the theoretical details in Supplementary Note 1) leads to the total evacuation of a valley in an attainable magnetic field. By shrinking the Fermi surface with Sb doping[28], one may lower $B_{empty}$ sufficiently to make field-induced 100% valley polarization as accessible as its spin counterpart.

## Methods

The dimensions of the samples used in this study were $2 \times 1 \times 1$ mm$^3$. They were cut by a wire-saw from a big crystal. The ratio of room-temperature to residual resistivity (RRR) was in the range of 30–50. Resistance was measured with a standard four-probe technique. A mechanical rotator was used to rotate the sample under pulsed magnetic field reaching 65 T. Gold wires 25 $\mu$m in diameter were attached to four electrodes on the sample using Dupont 4929NI silver paint. The angle could be determined from the signal obtained from a coil attached to the back of rotating platform, which monitored the magnitude of the field parallel to its axis. Electric current was applied along the trigonal direction. The results were reproduced in the study of two different samples and was repeated in two different laboratories (NHMFL-Los Alamos and WHMFC-Wuhan).

**Data availability.** The data that support the findings of this study are available from the corresponding author upon reasonable request.

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

## Acknowledgements

This work was supported by the 1000 Youth Talents Plan, the National Science Foundation of China (Grant No. 11574097), the National Key Research and Development Program of China (Grant No.2016YFA0401704). Z. Z. was also supported by directors' funding grant number 20120772 at LANL. R.D.M. acknowledges support from the US-DOE BES 'Science of 100T' program. The National High Magnetic Field Laboratory - PFF facility is funded by the National Science Foundation Cooperative Agreement Number DMR-1157490, the State of Florida and the U.S. Department of Energy. In France, this work was part of SUPERFIELD and QUANTUM LIMIT projects funded by *Agence Nationale de la Recherche*. B.F. acknowledges support from Jeunes Equipes de l'Institut de Physique du Collège de France (JEIP). K.B. was supported by China High-end foreign expert program, 111 Program and Fonds-ESPCI-Paris. Y.F. was supported by JSPS KAKENHI grants 16K05437, 15KK0155 and 15H02108.

## Author contributions

Z.Z. and K.B. designed the research project. Z.Z. and B.F. prepared the samples. Z.Z., J.W., H.Z. and R.D.M. carried out experiments. Y.F. provided theoretical support and calculations. Z.Z., Y.F. and K.B. wrote the manuscript and all authors commented on the manuscript.

## Additional information

**Competing interests:** The authors declare no competing financial interests.

**Publisher's note**: 

