## [Peer Review File · Nature Communications]

Reviewers' comments:

Reviewer #2 (Remarks to the Author):

The authors have adequately responded to the comments and recommendations made in my earlier report. In my opinion, the authors have also answered adequately the criticisms of the other reviewers.

Therefore, I recommend publication in Nature Communications.

Reviewer #4 (Remarks to the Author):

Review on the manuscript by Zengwei Zhu et al.

The manuscript by Zengwei Zhu et al presents a study on the electrical transport of bismuth crystal in magnetic field orders of magnitude higher than the quantum limit. Reading the manuscript and the first 3 reviewers' comments, as well as the replied from the authors, I reach the same conclusion by review 1 and review 3. I found that the study in the manuscript does not present a significant step compared with the authors' earlier work (Nature Physics 2012) as well as the manuscript by Miura et al (Physica B 1994).

Moreover, it is not clear to me why the drop of resistivity implies the empty of an electron valley. To the first order, wouldn't that imply the increase of carriers' density? In the supplement, the authors present a very complicated model to argue for a strong magnetic field dependence of the carrier mobility. To march the data, they need to present a non-monotonic field dependence of mobility. What is the physical origin? Is there any other experimental evidence for this assumption of field dependence of mobility and the empty of an electron valley, such as the Hall effect, or the thermoelectric measurements? We know that the authors' group is famous for the electrical and thermoelectric transport in intense fields. I am surprised that they did not try to bring other experimental evidence to support the rather complicated data analysis and the big claim in the manuscript.

In summary, I do not recommend the publication of the manuscript by Zhu et al. in Nature Communications.

Reviewer #5 (Remarks to the Author):

Zhu et al. report on the magnetic field dependent occupation of Dirac valleys in bismuth. They provide experimental results on the angular and temperature dependencies of the transverse magnetoresistance, which display a complete "drying of the Fermi sea" in one or two Dirac valleys for well-defined angles. A qualitative and a hand-waving quantitative explanation for the obtained results are given in a semi-classical transport theoretical picture.

Over almost a decade, the authors have progressively contributed to the fundamental understanding of the electronic properties of bismuth, among others. Based on their previous results, an angle-dependent study of the transverse magnetoresistance at very high magnetic fields was a logical step to take. Those challenging pulsed field magnetotransport experiments have confirmed what the authors must have expected before: a complete electronic depletion of one or more Dirac valleys at a sufficiently high magnetic field. The experimental data is novel, clear and indisputable, as well as the qualitative explanation, taking into account the anisotropic mobility per valley. I believe, however, that the manuscript's impact can benefit from a more elaborate quantitative explanation of the possible influence of Coulomb interactions between the

electrons and holes. As is frequently mentioned in the manuscript, such interactions possibly have a determining role in 'shaping the fine details'. Are the authors able to support these suggestions with more substantial information? For instance, what should be the occurrence of these interactions and their strength, in order to play a decisive role in the details of the experimental magnetoresistance data? And are these numbers reasonable to be achieved in this particular case?

I do believe this work deserves to be published in Nature Communications, as it uniquely reports and explains that Dirac valleys in bismuth can be fully emptied by means of a very high magnetic field. Obviously, this opens the way for investigations of related compounds under similar extreme conditions, in order to obtain additional knowledge about the fundamental physics underlying their electronic properties. When my questions related to a more in-depth explanation of the possible role of Coulomb interactions can be briefly addressed, as well as several typos and the wrongly cross-referenced Fig.3b be corrected, I recommend this manuscript for publication.

Reply to the referees:

Reviewer 4

I found that the study in the manuscript does not present a significant step compared with the authors' earlier work (Nature Physics 2012) as well as the manuscript by Miura et al (Physica B 1994).

The main result of the present work is the following: In presence of a sufficiently strong magnetic field all carriers of one or two out of the three electronic valleys leave that valley. There is no trace of such a statement in any of the two papers mentioned by the referee. The Nature Physics paper of 2012 reports that one can tune the contribution of different valleys to the total conductivity. The paper by Miura et al. identified a drop in magnetoresistance as another hole quantum oscillation. Therefore, if our conclusion is correct it is difficult to see how one can doubt that it presents a significant step forward.

Let us also highlight that this different conclusion is based on a more extensive set of data. We have performed angle-dependent magnetoresistance measurements up to 65 T. Along the high-symmetry axis, the data is extended up to 90.5 T and we include in the new version measurement of the Hall coefficient performed for the first time up to 60 T.

In the new version, we add the sentence in the concluding paragraph to stress this.

Moreover, it is not clear to me why the drop of resistivity implies the empty of an electron valley. To the first order, wouldn't that imply the increase of carriers' density?

This question, and more generally the often non-trivial connection between changes in resistivity and the underlying changes and carrier density and scattering rate, are at the heart of the quantum limit problem. Before answering this question, we invite the referee to carefully examine Figure 2b and 2e, which represent the theoretical carrier density as a function of magnetic field. This theoretical calculation, like the one performed as early as 1964 (See Fig. 10 in Phys. Rev. 135, A1118 (1964)) is backed by experimental evidence – without invoking this field-induced enhancement in carrier concentration one cannot attain a quantitative agreement with experimental Landau spectrum as reported in detail in earlier publications (see PNAS 109, 14813 (2012)). The fundamental reason behind this field-induced change in carrier density is simply the response of the system to maintain charge neutrality with the lowest energy electronic configuration, which results in continuously shifting the chemical potential.

Now let us return to the specific question of what happens near 40 T? Experimentally there is a drop in magnetoresistance near 40 T, which by comparison with calculation in the two panels Figure 2b and 2e we associate with a drop in total carrier density and a transfer of carriers between electron valleys. Although in the simplest level of approximation this drop in resistance and total carrier density appear at odds, we have presented well established reasons for this behavior in bismuth. This is indeed not what is expected in a naive picture. We thank the referee for giving us the opportunity to clarify this important message of our paper to physicists interested in magnetoresistance of real metals near the quantum limit.

In the supplement, the authors present a very complicated model to argue for a strong magnetic field dependence of the carrier mobility.

There are indeed several scenarios presented in the supplementary information, because their detail is not necessary for the reader to grasp the main idea behind the interpretation put forward in the main text; The magnitude of magnetoresistance is set by carrier concentration and carrier mobility. But, it is known that mobility in the case of Dirac valleys in bismuth is not a scalar but a tensor. The components of mobility differ by orders of magnitude. The magnitude of magnetoresistance is set by the relevant component (s) of mobility, which depends on the orientation of magnetic field. This is the fundamental reason behind angular oscillations of the magnetic field. Now, it is easy to understand how magnetoresistance can drop after emptying a valley. Consider a valley with a large contribution to the total magnetoresistance because of the amplitude of the relevant component of mobility. If the electrons in this valley leave it to join electrons in other valleys then magnetoresistance will drop. This is all one needs to provide a qualitative explanation for the drop in magnetoresistance. Following the reviewers' comment, we have inserted in the main text two simple equations (which were previously presented and discussed in the supplement) to show how intervalley transfer can give rise to a drop of magnetoresistance despite a drop in carrier concentration.

To match the data, they need to present a non-monotonic field dependence of mobility. What is the physical origin?

Our main conclusion is that the resistivity drop is concomitant with a valley becoming empty. As stated above, it does not need any field-dependence of mobility (either monotonic or non-monotonic). But the referee is right. A quantitative agreement requires a sudden change in the amplitude of mobility components. Its physical origin would be excitons, frequently invoked in this context and briefly discussed in our paper. However, this is not the main message of our paper.

Is there any other experimental evidence for this assumption of field dependence of mobility?

Magnetoresistance in bismuth is not B^2 . Therefore, it is natural to assume that mobility (or if you prefer scattering time) is a decreasing function of magnetic field.

Is there any other experimental evidence for this assumption of field dependence of mobility and the emptying of an electron valley, such as the Hall effect, or the thermoelectric measurements? We know that the authors' group is famous for the electrical and thermoelectric transport in intense fields. I am surprised that they did not try to bring other experimental evidence to support the rather complicated data analysis and the big claim in the manuscript.

We recall to the referee that we present a study performed at magnetic field intensities only available for a short duration. It is very difficult to do thermoelectric measurements in these pulsed field environments. Our previous high-field studies of thermoelectric response were performed in DC fields. The highest DC field available world-wide is 45 T and too low for the subject study of this paper. This is not the case of the Hall effect which we have performed in this field range and are including in the revised manuscript (See Fig. 4). As discussed in the new version, the Hall data confirm the main conclusion of the paper, namely, the total depletion of one or two electron valleys.

Reviewer 5

I believe, however, that the manuscript's impact can benefit from a more elaborate quantitative explanation of the possible influence of Coulomb interactions between the electrons and holes. As is

frequently mentioned in the manuscript, such interactions possibly have a determining role in 'shaping the fine details'.

The reviewer is right. We tried to quantify what we mean by "fine details". As seen in Fig. 4 of the supplement, in order to find a perfect match between semi-classical theory and experiment, one needs to invoke a two-fold drop either in μ_1 or in μ_2 . We have explicitly mentioned this in the new version.

Are the authors able to support these suggestions with more substantial information? For instance, what should be the occurrence of these interactions and their strength, in order to play a decisive role in the details of the experimental magnetoresistance data? And are these numbers reasonable to be achieved in this particular case?

At 50 T, the carrier density is around 10^{18}cm^{-3} . Therefore, the average electron-hole distance is about 10 nm. Therefore, assuming a dielectric coefficient of 100, the attractive Coulomb interaction between electrons and holes is estimated to be 0.7 meV. Excitonic effects become important if the average energy separation between an electron Landau level, a hole Landau level and the chemical potential becomes comparable to this. Following referee's recommendation we have inserted a new figure and a short discussion showing that this indeed happens. We thank the referee for this very helpful comment.

REVIEWERS' COMMENTS:

Reviewer #4 (Remarks to the Author):

I appreciate the authors' effort to address my concerns and others' comments in their revision. The inclusion of the 60 T Hall effect data is very helpful. In principle, the manuscript would meet the standard to Nature Communications. One part I think a further revision would help is to model the 60 T Hall effect data based on the fitting model presented in the supplement. So far the interpretation for the Hall effect in is simply contrasting B ~ 60 T trend of the Hall signal in two magnetic field orientations. The manuscript presents a quite complicated analysis. For such a multi-parameter fitting, it would be much convincing to demonstrate that the fitting scheme works not only in magnetoresistance but also in the Hall effect.

Reviewer #5 (Remarks to the Author):

In my opinion, the authors have adequately replied to the questions, comments and recommendations made by me and the other referees. There are still several typos left both in the article manuscript and the supplemental material, which can probably be corrected in the upcoming editorial process. I therefore recommend this manuscript for publication in Nature Communications.

Reply to the referees:

Reviewer 4

a further revision would help is to model the 60 T Hall effect data based on the fitting model presented in the supplement. So far the interpretation for the Hall effect in is simply contrasting $B \sim 60$ T trend of the Hall signal in two magnetic field orientations. The manuscript presents a quite complicated analysis. For such a multi-parameter fitting, it would be much convincing to demonstrate that the fitting scheme works not only in magnetoresistance but also in the Hall effect.

Response: As we have written in our text: “In semi-classical picture of multi-valley transport [24], the Hall conductivity is expected to vanish in the high-field limit ($\mu B \gg 1$) when the current is injected along symmetry axes.” In other words, the model we have used to treat magnetoresistance is not sufficient to explain the experimental observation of a Hall signal at high-field. As we have written in our text, this is because, “this picture assumes that carriers are not scattered from one valley to another. The finite Hall conductivity observed here indicates the existence of such scattering events.” Therefore, our treatment of the Hall signal is qualitative (i.e., the Hall signal vanishes when the system becomes single valley), in contrast to our quantitative treatment of magnetoresistance. We stress that even in the latter case, we are not fitting the data, aware that Aubrey’s treatment of multi-valley conductivity (ref. 24) is not complete.